# Zoomer: Enhancing MLLM Performance with Adaptive Image Focus Optimization

## Abstract

Recent advancements in multimodal large language models (MLLMs) have broadened the scope of vision-language tasks, excelling in applications like image captioning and interactive question-answering. However, these models struggle with accurately processing visual data, particularly in tasks requiring precise object recognition and fine visual details. Stringent token limits often result in the omission of critical information, hampering performance. To address these limitations, we introduce Zoomer, a novel visual prompting mechanism designed to enhance MLLM performance while preserving essential visual details within token limits. Zoomer features three key innovations: a prompt-aware strategy that dynamically highlights relevant image regions, a spatial-preserving orchestration schema that maintains object integrity, and a budget-aware prompting method that balances global context with crucial visual details. Comprehensive evaluations across multiple datasets demonstrate that Zoomer consistently outperforms baseline methods, achieving up to a 26.9% improvement in accuracy while significantly reducing token consumption.

## 1 Introduction

Recent advancements in multimodal large language models (MLLMs), such as GPT-4o, Gemini Pro, and Claude 3, have significantly expanded the capabilities of vision-language tasks. These models now excel in applications such as image captioning, object recognition, and interactive question-answering systems (Li et al., 2024; Gu et al., 2024). These models are able to process and integrate both text and images, creating opportunities for applications spanning diverse fields from creative writing to technical problem-solving. Yet, their reliance on textual prompts limits their performance when tasked with precise object recognition and interpreting intricate visual details (Cui et al., 2024). This limitation becomes evident when users attempt to guide the model using text prompts alone, as shown in Figure 1. While these models possess the capability to capture and analyze subtle visual nuances, current text-based prompting methods often fail to fully unlock this potential. We argue that more sophisticated approaches, where vision is not only analyzed but also serves as part of the prompt, are essential for refining the interaction and fully harnessing the power of MLLMs in high-stakes visual environments.

One of the primary limitations of current MLLMs arises from their unified image processing strategy. In visual tasks, there are often regions of interest (RoI) corresponding to the parts of concern and auxiliary understanding information, sometimes even including irrelevant and redundant background. However, most models process images uniformly without selectively filtering these regions. This lack of targeted processing leads to decreased fidelity in critical areas, severely affecting tasks that rely on small object detection or fine visual details. Although some recent studies (Jin et al., 2023) attempt to address this issue by modifying the model architecture, picking the valuable token and fine-tuning training, such approaches are impractical for black-box MLLMs.

Meanwhile, black-box MLLMs impose strict token limits on both text and visual prompts to ensure computational efficiency and user fairness (Chen et al., 2024). These constraints further compound the problem, as downscaling high-resolution images to fit within token budgets leads to the omission of vital details. As demonstrated in Figure 2, when high-resolution images are resized to meet the token limits imposed by these models, vital visual information is lost. In critical domains such as medical imaging or satellite-based analysis, missing such details can result in inaccurate or even dangerous outcomes (Mahapatra et al., 2022). Moreover, the black-box nature of these models limits

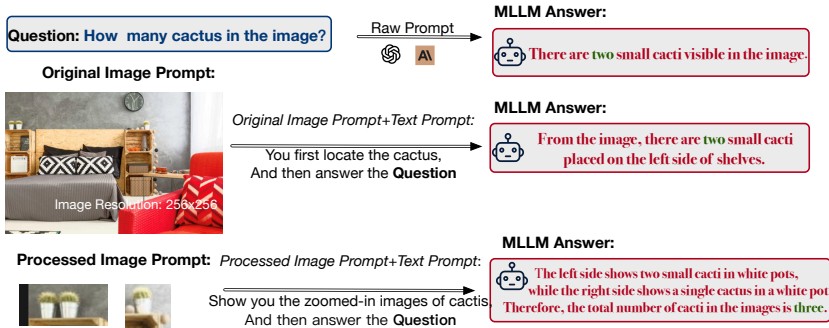

Figure 1: Illustration of a black-box MLLM's approach to counting cacti in an image. The model identifies two small cacti on the left side and overlooks the single cactus on the right side of the image, arriving at a total of three cacti. The processed prompt highlights specific regions of interest to facilitate the correct object count.

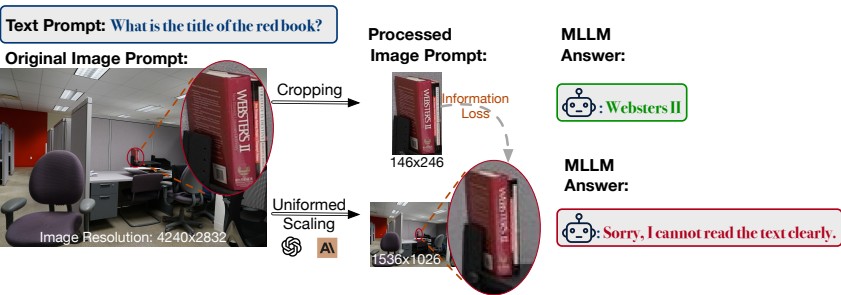

Figure 2: Illustration of information loss during image processing in black-box MLLMs. The original high-resolution image (4240x2832) is downscaled to meet token limits (1536x1026), leading to the loss of critical details. Cropping to focus on a region of interest (146x246) allows the model to correctly identify the book title as "Webster's II".

the ability to fine-tune or modify their architectures, making it difficult to address these shortcomings with conventional methods discussed in §3.1.

To tackle these challenges, we propose Zoomer, a novel visual prompting mechanism designed to preserve critical visual details while adhering to the token constraints of black-box MLLMs. Zoomer introduces three key innovations: (1) a prompt-aware visual emphasizing strategy that dynamically highlights the most contextually relevant parts of an image based on the input prompt such as shown in Figure 1 and 2, mitigating the information loss that typically results from resizing; (2) a spatial-preserving orchestration schema that maintains the structural integrity and relative positioning of objects within the image, thereby enhancing context-aware visual processing; and (3) a budget-aware prompting strategy that balances the need to capture global image context while preserving high-resolution slices, ensuring key visual details are retained without exceeding token budgets.

In a comprehensive evaluation across datasets such as *Vstar* (Wu & Xie, 2023), *CVBench* (Tong et al., 2024a), and *RealworldQA* (xAI, 2024), Zoomer consistently outperformed baseline methods. Notably, in the Vstar dataset, Zoomer-Patches achieved a 26.9% accuracy improvement over the baseline, while in RealWorldQA, Zoomer-Adaptive outperformed the baseline by 12.1%. Alongside these accuracy gains, Zoomer significantly reduced token usage. For example, on the TerraIncognita dataset, Zoomer achieved 6.4% higher accuracy with a 67% reduction in token consumption compared to the baseline. These results confirm that Zoomer not only addresses the limitations of visual processing in black-box MLLMs but also enhances efficiency. Moreover, across APIs like GPT-4o[1], Gemini-1.5Pro[2], and Claude-3.5-Sonnet[3], Zoomer demonstrated consistent improvements

---

[1] https://platform.openai.com/

[2] https://gemini.google.com/

[3] https://anthropic.com/

in both accuracy and token efficiency, solidifying its potential to optimize MLLM performance in real-world, high-resolution applications.

The contribution of this work can be summarized as follows: (1) We conduct a detailed investigation of GPT-4o's image prompting strategy, exposing key limitations in its handling of visual inputs. (2) We introduce Zoomer, a novel mechanism that addresses the challenge of preserving visual detail in black-box MLLMs while adhering to token constraints. (3) We present extensive experimental results across multiple datasets, demonstrating that Zoomer achieves significant improvements in both accuracy and token efficiency, offering valuable insights into enhancing multimodal processing in constrained environments.

## 2 PILOT EXPERIMENTS

One of the primary challenges faced by black-box MLLMs is their inability to process visual inputs efficiently, leading to diminished accuracy in fine-grained visual tasks. Models like GPT-4o often struggle with recognizing detailed or occluded objects, particularly when dealing with complex images. These limitations are compounded by token constraints, which restrict the amount of image data that can be processed in a single prompt. Furthermore, according to the Vision pricing calculator[4], GPT handle images by resizing and splitting them into basic units of 512×512 pixels. Each of these units corresponds to 170 tokens. This

| Method | Accuracy | Prompt Tokens |
|---|---|---|
| Unaltered Input | 0.57 | 955 |
| Image Crop | 0.58 | 270 |
| Zoomed Crop | 0.64 | 270 |

Table 1: Performance of different methods on Vstar-Bench Image Prompts.

method of processing not only imposes a strict limit on the image resolution but also increases the computational overhead due to the additional tokens generated from splitting. As a result, vital visual details may be lost when images are downsampled or resized to fit within the token limits, leading to poor performance on tasks that require precise visual grounding.

To evaluate this issue, we conducted a series of pilot experiments using GPT-4o-0513 on the *Vstar-Bench* dataset. This dataset challenges MLLMs to accurately identify detailed objects within high-resolution images, making it an ideal test for the model's capacity to handle fine-grained visual information. The experiments compared three different image processing strategies: (1) an unprocessed prompt (**Unaltered Input**), where the image is fed to the model in its original form; (2) a prompt where the image is cropped to focus on the target object (**Image Crop**); and (3) a prompt where the cropped image is further enlarged to emphasize the most relevant visual features (**Zoomed Crop**). Both the Image Crop and Zoomed Crop methods were constrained to fit within GPT-4o's patch size limit of 512x512 pixels.

As shown in Table 1, the **Zoomed Crop** method significantly outperformed the others, achieving an accuracy of 0.76 with a token usage of 270. In comparison, the **Unaltered Input** method, despite processing the entire image, only achieved an accuracy of 0.64 while consuming 955 tokens. Similarly, the **Image Crop** method, although reducing the token count to 270, did not yield any improvement in accuracy compared to the unprocessed input.

These results highlight a fundamental problem in current black-box MLLMs: they fail to efficiently manage the trade-off between image resolution and token constraints. In cases like those presented by the *Vstar-Bench* dataset, where fine-grained visual information is critical, processing unaltered high-resolution images leads to excessive token consumption without improving accuracy. While the **Image Crop** method reduces token usage, it fails to improve performance because simply cropping an image without emphasizing key details does not provide sufficient context for the model to interpret the visual input accurately.

The superior performance of the **Zoomed Crop** method underscores the importance of vision enhancement techniques in black-box MLLMs. By focusing on the most relevant portions of the image, **Zoomed Crop** preserves critical details while remaining within token limits, enabling the model to interpret detailed visual inputs more effectively. This approach resolves a common issue faced by black-box MLLMs, where downscaling or cropping images to meet token requirements often leads to a loss of essential information, reducing the overall effectiveness of the model.

---

[4]https://openai.com/api/pricing/

Our experiments reveal that without adaptive techniques like **Zoomed Crop**, black-box MLLMs struggle to process high-resolution images efficiently, limiting their performance on tasks that require precise visual recognition. These findings demonstrate the necessity of vision enhancement strategies to address the inherent limitations of token-constrained MLLMs.

## 3 RELATED WORK

### 3.1 MULTIMODAL LLMS: OPEN-SOURCE AND BLACK-BOX MODELS

The integration of visual and textual modalities in large language models (LLMs) has led to significant advancements in multimodal models (MLLMs) like GPT-4o, Gemini Pro and Claude3-Sonnet. These models rely on effective visual encoding strategies to bridge the gap between language and vision. Approaches such as CLIP (Yang et al.) align visual and language embeddings through contrastive learning, while models like Flamingo (Alayrac et al.) and BLIP-2 (Dai et al.) use cross-attention mechanisms or pretraining modules to link vision encoders with LLMs. However, these methods often rely on fixed low-resolution inputs (e.g., 224x224), limiting their ability to process high-resolution images or non-standard aspect ratios (Liu et al., a), which hampers performance on fine-grained tasks such as OCR and small object detection

In contrast, open-source multimodal models (Li et al., c; Xu et al.; Zhang et al., a; Li et al., a; Zhao et al.) allow for architectural modifications and fine-tuning to accommodate any-resolution inputs. However, black-box MLLMs such as GPT-4o and Gemini Pro, which impose strict token limits for computational efficiency, require alternative solutions. The need to downsample or crop images to meet these constraints often results in the loss of crucial visual details, particularly in tasks requiring detailed visual understanding. While position embedding interpolation (Bai et al.; Wang et al.; Luo et al.; Hong et al.; Chen et al.) and patch-based cropping (Xu et al.; Li et al., a) widely adpoted in open-soure models offer promising directions for any aspect ratio and any-resolution image processing, they are not applicable to black-box models, where architectural changes and extra training/fine-tuning are not permitted.

### 3.2 OBJECT DETECTION

Traditional object detection models, such as Faster R-CNN (Ren et al.) and YOLO (Redmon et al.), effectively identify and localize objects within predefined categories. However, they struggle with open-set scenarios, where novel objects not seen during training need to be detected.

Recent advances address this limitation through open-set detection models that leverage natural language processing. For instance, OV-DETR (Zang et al.) integrates CLIP with object detection to generate category-specific bounding boxes from textual prompts, enabling detection in open-world settings. Similarly, GLIP (Li et al., b) reframes detection as a grounding problem, improving alignment between visual regions and textual descriptions. DetCLIP (Yao et al.) extends this further using pseudo labels from large-scale captioning datasets, enhancing generalization. Grounding DINO (Liu et al., b), built on the DETR framework (Carion et al.), also advances open-set detection through natural language integration.

In addition, SAM (Kirillov et al., 2023) and SAM-2 (Ravi et al., 2024) offer zero-prompt or minimal-prompt segmentation for arbitrary objects but lack robust text-prompt handling. EVF-SAM (Zhang et al., b) overcomes this by extending SAM's capabilities to better manage complex text-based object segmentation.

By incorporating these models, Zoomer enhances its ability to dynamically detect and emphasize regions of interest (RoIs), enabling black-box MLLMs to focus on the most relevant visual content without losing critical details, which is essential for maintaining high performance across varied resolutions.

## 4 METHOD OVERVIEW

Inspired by the observation derived from our pilot experiments, we propose Zoomer, a comprehensive visual prompting mechanism designed to effectively address the loss of detail in images that occurs during the naive resizing process in current black box multimodal LLMs, such as GPT-4

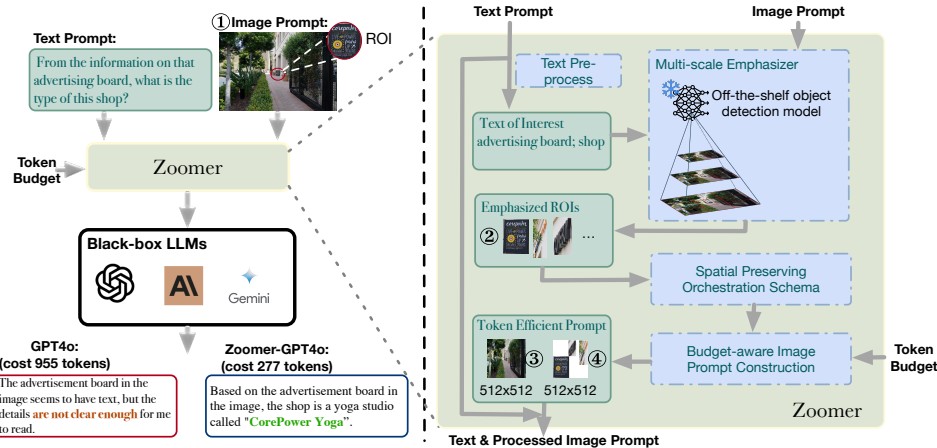

Figure 3: The Zoomer framework. Left: Raw Input image (①) and text prompt are processed by Zoomer and then fed into a black-box LLM (e.g., GPT-4o) for analysis, resulting in more accurate and detailed responses compared to standard input methods with even token saving. Right: Zoomer processes the text to extract key terms and uses a multi-scale emphasizer(§4.1) with an off-the-shelf object detection model to identify regions of interest (ROIs). The identified ROIs (②) are then processed through a spatial preserving orchestration schema (§4.2) for a filtered emphasized patch (④) and a budget-aware image prompt construction module (§4.3) to create a token-efficient prompt within the specified budget. A scaled global view (③) is also generated for potential prompting.

and Gemini 1.5. As illustrated in Figure 3, our mechanism comprises three key components: (1) A prompt-aware visual emphasizer that allocates high-fidelity image slices based on prompt texts to facilitate efficient and focused visual encoding; (2) A spatial-preserving encoding schema that consolidates the collected image slices while maintaining their relative spatial positions to create a condensed visual input; (3) A budget-aware prompting strategy that maximizes the accuracy of results obtained from the black box models while fits the budget requirement from users.

## 4.1 Prompt-aware Visual Emphasizer

The prompt-aware visual emphasizer utilizes a multi-scale emphasizing strategy to prioritize image slices that are most relevant to the input prompts. By analyzing the semantic content of the prompts, this component dynamically selects and enhances specific regions of the image at varying resolutions. This approach not only enriches the contextual information available to the model but also mitigates the adverse effects of losing critical details during the resizing process.

**Prompt Tokenization** Prompt tokenization is a critical first step in which input prompts are parsed into meaningful tokens. This process segments the prompt into components that can be easily analyzed for semantic relevance. Specifically, the prompt is divided into structural components, and our focus is on processing the relevant sections that contribute directly to visual emphasis.

To enhance the extraction of semantically relevant tokens, we apply advanced natural language processing (NLP) techniques. First, we use the NLTK library[5] to remove stopwords, reducing noise and ensuring that the model's attention remains on the most critical visual elements. By eliminating these non-essential words, we concentrate on key terms that directly influence the visual emphasis.

In addition to basic stopword removal, we utilize dependency parsing (Sarthi et al., 2024; De Marneffe & Manning, 2008) to analyze the syntactic structure of the prompt. This deeper analysis identifies core entities and relationships, such as subject-object pairs and action verbs, which are crucial for interpreting the user's intent. By focusing on these core semantic elements, we ensure that the visual emphasis aligns precisely with the underlying meaning of the prompt.

---

[5]https://www.nltk.org/

Finally, we strip away any irrelevant formatting or non-content-related details, allowing the visual emphatizer to focus solely on the essential information. This multi-layered tokenization approach ensures an optimal match between the tokenized prompt and the image features selected for emphasis.

**Multi-Scale Emphasizing Algorithm:** Given a key object term extracted from the text prompt, the Multi-Scale Emphasizing Algorithm 1 utilizes a state-of-the-art object detection model to localize the corresponding object in the image prompt. In our experiments, we primarily employ GroundingDINO (Liu et al., b) as our localization model.

The encoder in such models typically downsamples the input image to a resolution of $224 \times 224$ or $336 \times 336$, potentially resulting in information loss when localizing the target object at a coarse granularity. To address this limitation, we propose a Multi-Scale Emphasizing Algorithm that processes the original image at multiple resolutions. The algorithm divides the input image into patches at various granularities, *e.g.,* $2 \times 2$, $3 \times 3$, and beyond. For each generated patch, we apply the object detection model to localize the target object. The algorithm retains bounding boxes returned by the model that exceed a predefined confidence threshold. These high-confidence bounding boxes collectively form the output of our algorithm, providing a comprehensive multi-scale representation of the target object's location.

---

**Algorithm 1** Multi-Scale Emphasizing Algorithm

---

**Require:** $I$: input image, $k$: key object term, $M$: object detection model, $T$: confidence threshold
**Ensure:** $B$: set of bounding boxes
 1: $B \leftarrow \emptyset$
 2: $S \leftarrow \{2, 3, \ldots, S_{\max}\}$                                    ▷ Set of scaling factors
 3: **for** each $s \in S$ **do**
 4:      $P_s \leftarrow \texttt{DivideIntoPatches}(I, s \times s)$
 5:      **for** each patch $p \in P_s$ **do**
 6:          $b, c \leftarrow M(p, k)$                          ▷ Get bounding box and confidence
 7:          **if** $c \geq T$ **then**
 8:              $B \leftarrow B \cup \{b\}$
 9:          **end if**
10:      **end for**
11: **end for**
12: **return** $B$

---

## 4.2 Spatial-preserving Orchestration Schema

Building upon the Multi-Scale Emphasizing Algorithm, we introduce a Spatial-preserving Orchestration Schema to maintain the structural integrity of the image during the encoding process. This schema filters the bounding boxes obtained from the Multi-Scale Emphasizing Algorithm and ensures that the relative positions of the selected image slices are preserved, facilitating a more faithful representation of the original image layout and enabling coherent reconstruction when processed by the multimodal LLM. To refine the selection of bounding boxes, we implement a Non-Maximum Suppression (NMS) based slice filtering method. NMS is employed to eliminate redundant and overlapping slices, retaining only the most salient features that align with the prompt. The process works as described in Algorithm 2.

By setting an appropriate threshold $T$ for the Intersection of Union (IoU) of bounding boxes around the selected regions, we ensure that only the highest-quality slices are retained for the encoding process. This filtering step enhances computational efficiency by reducing the number of slices to be processed and improves the clarity and relevance of the visual information provided to subsequent stages of the model.

The resulting set of filtered slices are then orchestrated to preserve their original relative positions within the image. This orchestration process involves the following steps: **Slice Extraction**: For each bounding box $b_i$ in the filtered set $F$, we extract the corresponding image slice from the original image. **Blank Image Creation**: We create a new blank image with the same dimensions as the original image. **Slice Placement**: We place each extracted slice onto the blank image at its original position, leaving the rest of the image blank. **Image Shrinking**: The resulting image, con-

taining only the selected slices in their original positions with the rest left blank, is then shrunk to a predetermined size while maintaining its aspect ratio.

### 4.3 BUDGET-AWARE PROMPTING STRATEGY:

Our approach incorporates a sophisticated budget-aware prompting strategy that optimizes the allocation of token budget for image processing. This strategy begins with a user-specified total token budget $B_{total}$, allowing for customization based on specific task requirements or computational constraints. We propose four varieties of Zoomer to accommodate different budget scenarios and task requirements:

• Zoomer-**Local(④):** This variant utilizes only the spatial-preserving schema to consolidate all focused image slices into a single image patch(④ in Figure 3). It is optimal for scenarios with very limited token budgets, prioritizing the most relevant visual information.

• Zoomer-**Adaptive (④ + ◇ ③ ):** This approach dynamically includes a global view of the original image if the cropped portion falls below a certain threshold $T_A$. This allows the MLLM to better understand the overall scene context when the budget permits, while still focusing on key areas of interest.

• Zoomer-**Global (④ + ③):** This variant assigns a global view to all images, regardless of the specific regions of interest. It is suitable for tasks that require consistent overall context and when the token budget is sufficient to include both global and local information.

• Zoomer-**Patches(② + ③):** This is the most token-intensive approach, assigning each image slice its own patch without spatial preservation, along with a global view. It provides the most detailed information but requires the largest token budget.

The selection among these varieties depends on the user-specified budget and the nature of the task. For each variant, the number of high-resolution slices or patches $N$ is calculated based on the available budget and the token cost per slice or patch. These slices are selected from the output of our Multi-Scale Emphasizing Algorithm, prioritizing based on their relevance to the key term of text prompts. To present the methods more clearly and vividly, we refer to Figure 4, which outlines the methodology, and Figure 5, which showcases a specific case study.

## 5 EXPERIMENTS

In this section, we evaluate the performance of Zoomer through a series of experiments designed to test its ability to improve token efficiency and preserve visual fidelity across different black-box MLLM. Specifically, we aim to answer the following questions: **(i) Accuuracy**: Does Zoomer improve accuracy across different black-box MLLMs on image-related tasks? **(ii) Efficiency**: How does Zoomer perform compared to baseline methods in terms of both accuracy and token efficiency? **(iii) Component Contribution**: What is the impact of key components in Zoomer, such as multi-scale vision emphasize and the budget-aware prompt strategy?

### 5.1 SETUP

**Assessment and Datasets** We evaluated our system on a series of challenging multimodal tasks, using commercial black-box MLLMs for applications ranging from visual-language reasoning to image understanding and question answering. The experiments were conducted on a variety of different public datasets, including:

1) *Vstar* (Wu & Xie, 2023): A benchmark dataset focused on image classification, used to evaluate fine-grained visual recognition capabilities in object detection and classification tasks.

2) *CVBench* (Tong et al., 2024a): Contains 2 sub-category, $CVBench_{2D}$ and $CVBench_{3D}$, respectively, representing two-dimensional and three-dimensional visual image, respectively, to evaluate the performance of the model when processing images of different dimensions, especially the understanding ability in complex scenes.

3) *RealworldQA* (xAI, 2024): Used to test the multimodal question answering performance of the model in real-world scenarios, involving cross-language and cross-image information processing.

4) *MMVP* (Tong et al., 2024b): A validation set for multimodal visual processing, designed to evaluate the comprehensive understanding of models for complex visual scenes.

5) *ScienceQA* (Lu et al., 2022): A multimodal scientific question-answering dataset featuring multiple-choice questions across a diverse range of science topics.

**Models** We employed three black-box MLLMs—GPT-4o-0513, Claude-v3-Sonnet, and Gemini-Pro—accessed via their respective APIs (OpenAI, Claude, Google). Across all experiments, we set the temperature to 0 and used greedy decoding for consistency, optimizing the stability of outputs. NMS was applied with a confidence score threshold of 0.8 to filter irrelevant regions from high-resolution images.

**Metrics** We used classification accuracy across all examples as the primary evaluation metric. Additionally, we compared token usage for each model configuration to evaluate the efficiency improvements offered by Zoomer.

**Baselines** We compare Zoomer against the following baseline methods:

1) *Raw*: This baseline feeds MLLM the unmodified prompt, with no adjustments made to the image.

2) *Resize*: Here, images larger than 512x512 pixels are resized to fit within the GPT-4o's patch limit, while smaller images remain unchanged.

## 5.2 MAIN RESULTS

Table 2 compares the performance of Zoomer against baseline methods across various datasets using GPT-4o. The results show that Zoomer, particularly in its Patches and Adaptive versions, consistently outperformed baseline approaches in terms of accuracy while maintaining lower token usage than the Raw method. Zoomer consistently outperformed baseline methods, showing accuracy improvements up to 26% across multiple tasks. For example, in the *Vstar* dataset, Zoomer-Patches achieved an accuracy of 0.717, compared to 0.565 using the Raw baseline, marking a 26.9% improvement. In *RealworldQA*, which demands complex multimodal reasoning, Zoomer-Adaptive achieved 0.758 accuracy, outperforming the 0.676 accuracy of the Raw method by 12.1%. These results highlight that Zoomer effectively preserves fine-grained visual details, enabling improved object recognition and image understanding across real-world tasks, where precise detail retention is crucial.

We further evaluated Zoomer on Claude-3.5-Sonnet and Gemini-1.5Pro to assess its generalizability across different black-box MLLMs. Table 3 shows that Zoomer demonstrated robust performance across different black-box MLLMs, for example, Zoomer achieved an accuracy of 0.704 on Vstar, compared to 0.531 with the Raw baseline, marking a 32.6% improvement. Similarly, in Claude-3.5-Sonnet, Zoomer outperformed the baseline by 34.5% on RealworldQA, improving from 0.610 to 0.741. These results suggest that Zoomer can consistently enhance performance across different architectures, making it a versatile tool for various MLLM-based applications.

A key contribution of Zoomer is its ability to reduce token consumption while maintaining or improving accuracy. As shown in Table 4, Zoomer consistently delivers both token efficiency and performance improvements across various benchmark datasets. For instance, on the TerraIncognita dataset, guided by ManyICL (Jiang et al., 2024), Zoomer achieves 0.83 accuracy using 315 tokens, compared to the Raw baseline's 0.78 accuracy with 963 tokens—a 67% reduction in token usage while improving performance by 6.4%. Additionally, Zoomer reduces latency, making it practical for real-time applications. In the TerraIncognita zero-shot setting, Zoomer lowered latency from 4.8s to 3.1s, a 35.4% reduction without sacrificing accuracy. This makes Zoomer highly suitable for tasks like autonomous driving, and real-time visual analytics, where both token efficiency and reduced latency are critical.

## 5.3 FINDINGS

Here we analyze why the Zoomer-Patches version underperforms compared to Zoomer-Global and even the Local version on certain datasets. For example, on the $CVBench_{3D}$ dataset, the accuracy of the Patches version is 0.025 lower than the Global version and 0.04 lower than the Local version. Similarly, on the *MMVP* dataset, the Patches version falls short by 0.005 compared to

| Acc./Tokens ⟍ Bench Method | Vstar | CVBench-2D | CVBench-3D | RealworldQA | SQA-I | MMVP |
|---|---|---|---|---|---|---|
| Raw | 0.565/955 | 0.685/428 | 0.782/895 | 0.676/998 | 0.873/353 | 0.833 /270 |
| Resize | 0.419/270 | 0.663/270 | 0.752/270 | 0.611/270 | 0.868/270 | 0.833/270 |
| Zoomer-Local | 0.671/270 | 0.724/270 | 0.862/270 | 0.724/270 | 0.883/270 | 0.871/270 |
| Zoomer-Adaptive | 0.675/419 | 0.729/374 | 0.879/408 | 0.747/362 | 0.911/308 | 0.887/351 |
| Zoomer-Global | 0.676/540 | 0.731/540 | **0.883**/540 | 0.753/540 | 0.923/540 | **0.889**/540 |
| Zoomer-Patches | **0.717**/1029 | **0.746**/709 | 0.858/1113 | **0.758**/997 | **0.928**/727 | 0.884/726 |

Table 2: Performance of GPT-4o-0513 across different datasets using various image prompt processing methods, focusing on accuracy and token consumption. Among these approaches: **Local**: Only the extracted RoIs are used. **Adaptive**: Selectively provides the MLLM with a global view of the image based on the prompt strategy. **Global**: Every request includes the global view of the image. **Patches**: Does not use the Spatial-Preserving Orchestration Schema; instead, each possible RoI is independently provided to the MLLM, including the global view.

| API | Method | Vstar | CVBench-2D | RealworldQA | MMVP |
|---|---|---|---|---|---|
| GPT-4o | Raw | 0.565 | 0.685 | 0.676 | 0.833 |
| | Zoomer | 0.717 | 0.746 | 0.758 | 0.889 |
| Gemini-1.5Pro | Raw | 0.531 | 0.654 | 0.640 | 0.798 |
| | Zoomer | 0.704 | 0.732 | 0.739 | 0.878 |
| Claude-3.5-Sonnet | Raw | 0.518 | 0.667 | 0.610 | 0.802 |
| | Zoomer | 0.697 | 0.728 | 0.741 | 0.872 |

Table 3: Accuracy of Different Black-box MLLM APIs. For *Vstar*, *CVBench-2D*, and *RealworldQA*, we used the Patches version of SysName. For *MMVP*, inspired by Table 2, we employed the Global version.

the Global version and by 0.003 compared to the Adaptive version. Given that these results are averaged over multiple measurements, and accounting for model fluctuations, we hypothesize that this performance drop occurs because the Patches version treats each RoI as an independent image and provides them separately to the MLLM. When there are too many RoIs, the model may fail to capture or integrate some of them, leading to a drop in accuracy.

| Method | Accuracy | | Tokens | | Latency | | Money Cost($10-e3) | |
|---|---|---|---|---|---|---|---|---|
| | Zero-Shot | 15-Shot | Zero-Shot | 15-Shot | Zero-Shot | 15-Shot | Zero-Shot | 15-Shot |
| Raw | 0.78 | 0.84 | 963 | 13488 | 4.8s | 18.7s | 4.815 | 67.44 |
| Resize | 0.61 | 0.74 | 255 | 4080 | 2.9s | 7.5s | 1.275 | 20.4 |
| Low-Detail | 0.6 | 0.7 | 85 | 1360 | 2.1s | 6.5s | 0.425 | 6.8 |
| Zoomer-Adaptive | 0.83 | 0.88 | 315 | 5112 | 3.1s | 9.8s | 1.575 | 25.56 |

Table 4: Performance in terms of accuracy, latency, and image token cost on TerraIncognita under ICL conditions—specifically with 15 examples per question—and under zero-shot conditions.

## 5.4 Ablation study

To further understand the impact of the individual components within Zoomer, we conducted an ablation study focusing on two key variants: 1) Zoomer with multi-scale emphasize: Compared with the commonly used multi-resolution and directly use, multi-scale visual emphasis is used to identify and emphasize RoI in the image. 2) Zoomer with different models.

To further investigate the contributions of Zoomer's components, we conducted an ablation study (Table 5). The results demonstrate that the combination of multi-scale visual emphasis and Patches prompt strategies delivers the best performance across all most datasets. Comparing different vision emphasis models, such as EVF-SAM and Ground Dino, further highlights the effectiveness of Zoomer. Despite differences in model capabilities, both show accuracy improvements across datasets. Additionally, when comparing different emphasis methods—Default, Multi-Resolution, and Multi-Scale—the Multi-Scale method consistently outperformed Multi-Resolution. We hypoth-

| Emphasize Method | Model | Prompt Strategy | VSTAR | CVBench-2D | CVBench-3D | RealworldQA | SQA_I | MMVP |
|---|---|---|---|---|---|---|---|---|
| Default | EVF-SAM | Local | 0.571 | 0.676 | 0.799 | 0.685 | 0.878 | 0.840 |
| | | Adaptive | 0.575 | 0.713 | 0.825 | 0.721 | 0.883 | 0.853 |
| | | Global | 0.578 | 0.721 | 0.830 | 0.724 | 0.888 | 0.873 |
| | | Patches | 0.571 | 0.727 | 0.838 | 0.721 | 0.868 | 0.847 |
| | Ground Dino | Local | 0.581 | 0.706 | 0.825 | 0.715 | 0.853 | 0.843 |
| | | Adaptive | 0.583 | 0.715 | 0.839 | 0.731 | 0.901 | 0.856 |
| | | Global | 0.583 | 0.718 | 0.848 | 0.734 | 0.903 | 0.878 |
| | | Patches | 0.588 | 0.706 | 0.831 | 0.726 | 0.909 | 0.877 |
| Multi-Resolution | EVF-SAM | Local | 0.584 | 0.692 | 0.840 | 0.725 | 0.883 | 0.847 |
| | | Adaptive | 0.585 | 0.713 | 0.853 | 0.731 | 0.911 | 0.868 |
| | | Global | 0.584 | 0.721 | 0.858 | 0.734 | 0.918 | 0.878 |
| | | Patches | 0.602 | 0.713 | 0.857 | 0.731 | 0.913 | 0.868 |
| | Ground Dino | Local | 0.636 | 0.718 | 0.826 | 0.702 | 0.888 | 0.851 |
| | | Adaptive | 0.644 | 0.718 | 0.843 | 0.706 | 0.903 | 0.865 |
| | | Global | 0.664 | 0.722 | 0.847 | 0.714 | 0.918 | 0.869 |
| | | Patches | 0.662 | 0.726 | 0.836 | 0.702 | 0.921 | 0.869 |
| Multi-Scale | EVF-SAM | Local | 0.637 | 0.721 | 0.852 | 0.704 | 0.853 | 0.857 |
| | | Adaptive | 0.643 | 0.724 | 0.861 | 0.706 | 0.901 | 0.876 |
| | | Global | 0.643 | 0.728 | 0.879 | 0.717 | 0.903 | 0.888 |
| | | Patches | 0.672 | 0.737 | 0.871 | 0.730 | 0.909 | 0.880 |
| | Ground Dino | Local | 0.671 | 0.724 | 0.862 | 0.724 | 0.883 | 0.871 |
| | | Adaptive | 0.675 | 0.729 | 0.879 | 0.747 | 0.911 | 0.887 |
| | | Global | 0.676 | 0.731 | 0.883 | 0.753 | 0.923 | 0.889 |
| | | Patches | 0.717 | 0.746 | 0.858 | 0.758 | 0.928 | 0.884 |

Table 5: Performance of Zoomer Across Datasets for Different Emphasis Methods, Models, and Prompt Strategies.

esize that, while Multi-Scale crops images and may split objects, its pyramid-shaped multi-recall strategy compensates for this by enhancing recall. In contrast, although Multi-Resolution maintains object integrity, adjusting resolution disrupts the model's performance, likely because most models are trained on fixed-size inputs, and changing the resolution weakens their inherent capabilities.

# 6 CONCLUSION

In this paper, we introduced Zoomer, a novel visual prompting mechanism designed to overcome the limitations of black-box MLLMs in processing images while adhering to token constraints. Our approach effectively balances the need to capture essential visual details without exceeding token budgets, a challenge commonly encountered in existing models like GPT-4o and Gemini Pro.

Through a comprehensive evaluation across datasets such as Vstar and RealWorldQA, Zoomer demonstrated significant improvements, particularly in fine-grained visual tasks. Our results show that Zoomer-Patches achieved a 26.9% accuracy gain over baseline methods in Vstar, and Zoomer-Adaptive provided a 12.1% improvement in RealWorldQA. These gains were achieved while drastically reducing token usage, with Zoomer delivering 6.4% higher accuracy in the TerraIncognita dataset using 67% fewer tokens.

Although this work primarily focuses on improving the efficiency of visual processing in black-box MLLMs, another potential issue that arises in real-world applications is communication cost. For example, transferring large images from edge devices (e.g., wearable cameras or glasses) to cloud servers can be expensive in terms of bandwidth, latency, and energy consumption. While Zoomer is designed to reduce token usage, its application in minimizing data transmission costs is an area that could be explored in future work.

As part of future research, we plan to investigate how Zoomer could be adapted for edge ML applications, enabling local processing on devices such as wearable cameras. This would allow for more efficient handling of visual inputs at the edge, reducing the need for extensive data transfers to the cloud. We aim to measure latency, power consumption, and the overall impact on system performance to assess the feasibility of applying Zoomer in these scenarios.

In summary, Zoomer offers a practical solution for enhancing visual processing in constrained MLLMs and opens up new directions for future exploration.

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

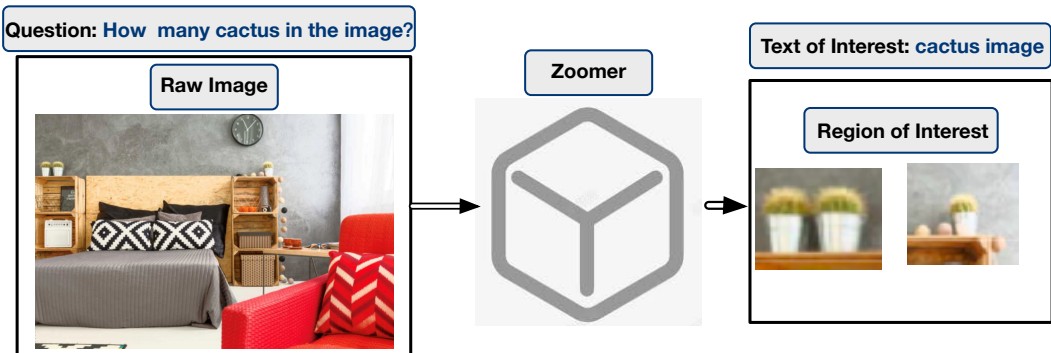

Figure 4: The example of applying Zoomer

# A APPENDIX

## A.1 DETAILS OF THE METHOD

Figure 4 is an example of the Zoomer, and Figure 5 is the output of various versions of the Zoomer.

## A.2 ALGORITHM OF THE NMS

---
**Algorithm 2** NMS-based Slice Filtering
---
**Require:** $B$: set of bounding boxes, $T$: IoU threshold
**Ensure:** $F$: set of filtered bounding boxes
1:  $F \leftarrow \emptyset$
2:  Sort $B$ in descending order of confidence scores
3:  **while** $B \neq \emptyset$ **do**
4:      $b_{\max} \leftarrow \arg\max_{b \in B} \text{score}(b)$
5:      $F \leftarrow F \cup b_{\max}$
6:      $B \leftarrow B \setminus b_{\max}$
7:      **for** each $b \in B$ **do**
8:          **if** $\text{IoU}(b_{\max}, b) \geq T$ **then**
9:              $B \leftarrow B \setminus b$
10:         **end if**
11:     **end for**
12: **end while**
13: **return** $F$
---

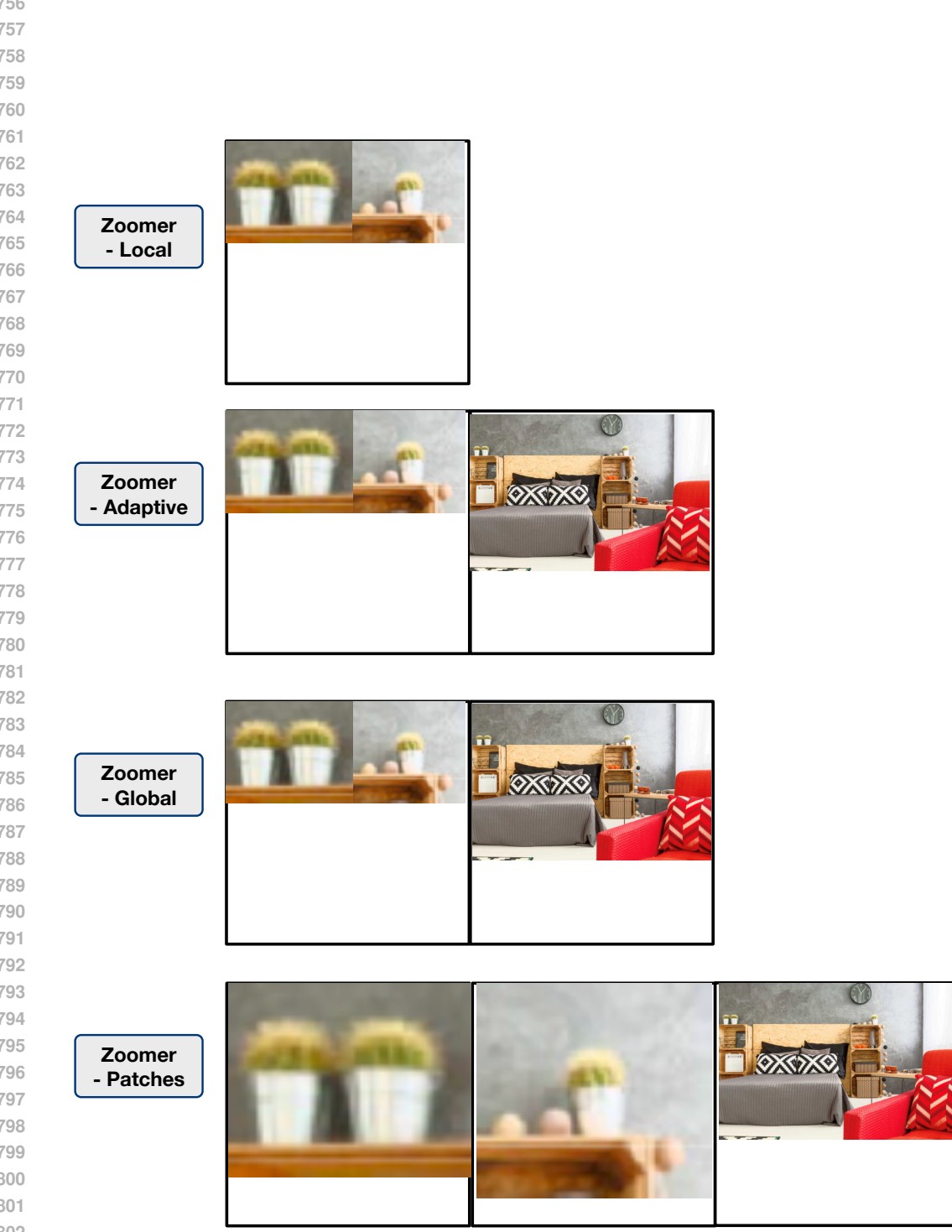

Figure 5: The example of different settings of Zoomer

