# OpenReview forum: "Zoomer: Enhancing MLLM Performance with Adaptive Image Focus Optimization"
_ICLR.cc/2025/Conference — ICLR 2025 Conference Withdrawn Submission_

### Official Review · Reviewer_7KgM · 2024-10-18

**Soundness:** 1
**Presentation:** 2
**Contribution:** 1
**Rating:** 3
**Confidence:** 4

**Summary:**

This paper proposes Zoomer, a test-time technique to save token inputs for proprietary multimodal large language models, such as GPT-4o. Given a high-resolution image, Zoomer uses an off-the-shelf object detector, GroundingDINO, to predict region proposals, which are cropped and fed into MLLMs. This paper uses proprietary models with raw visual resolutions as baselines, and shows performance gains on five recent benchmarks.

**Strengths:**

**Motivation is straightforward**. A large portion of existing MLLMs resize high-resolution images to low-resolution (e.g., 336 for LLaVA), so that visual token sequence length will not exceed context length limit of large language model. A straightforward technique is to crop needed regions and get rid of what may not be needed before feed it into MLLMs.

**Sufficient empirical studies on proprietary models**. This paper takes GPT-4o, Claude, Gemini as its baselines to support its claim, and include sufficient studies on object detection backbones, visual feature tricks (like multi-resolution, multi-scale inputs).

**Weaknesses:**

**Lack of novelty.** The proposed method in this paper feel more like an engineering effort on API supported models. It heavily relies on a strong object detector such as GroundingDINO, without unique contributions or novel insights. I cannot see a major architectural modifications with comparison to GroundingDINO. This paper emphasizes tool use of GroundingDINO and proprietary GPT, which could be a technical extension to GroundingDINO. But I do not think this has a major difference to GroundingDINO.

**Limited evaluations.** Considerable amount of open-sourced efforts have studied on designing high-resolution-input MLLMs, such as [A, B]. This paper does not include sufficient comparisons and introductions over these works, and claim clearly about differences between Zoomer and these models. More meticulous comparisons are needed before publication of this paper.

[A] Mini-Gemini: Mining the Potential of Multi-modality Vision Language Models.

[B] InternLM-XComposer2-4KHD: A Pioneering Large Vision-Language Model Handling Resolutions from 336 Pixels to 4K HD.

**Questions:**

No questions so far.

---

### Official Review · Reviewer_eN5G · 2024-11-03

**Soundness:** 3
**Presentation:** 3
**Contribution:** 3
**Rating:** 6
**Confidence:** 4

**Summary:**

This paper mainly focuses on enhancing the recognition and understanding of fine-grained objects in proprietary MLLMs. The authors propose an approach where, based on the user query, relevant regions are first detected using methods like Grounding-DINO. These detected regions are then organized to be fed to the MLLM as additional visual prompts to enhance its comprehension. Experiments were conducted on multiple closed-source models to validate the effectiveness of this approach.

**Strengths:**

Fine-grained recognition and understanding are crucial capabilities for MLLMs. This paper makes a compelling case for using external tools to provide models with additional detailed information, enhancing their comprehension.

**Weaknesses:**

1. The image shrinking process described in L323-325 is unclear. Does "shrink" refer to resizing the image? If so, will the cropped image still appear blurred or vague after the resize operation, given that the blank image is created at the same dimensions as the original image (L322)? If not, how does this shrinking process ensure that the processed image will fit within a 512*512 size? It is suggested to include a step-by-step description for the shrink process.
2. The accuracy values mentioned in L144-147 do not match the data presented in Table 1. Specifically, Zoomed Crop and Unaltered Input are argued to achieve 0.76 and 0.64 accuracy respectively, but in Table 1, the reported numbers are 0.64 and 0.57 respectively.
3. The symbol T is used inconsistently across the paper. In Algorithm 1, T represents the confidence threshold, while in Algorithm 2, it denotes the IoU threshold.

**Questions:**

Overall is good to me, it would be better to provide more details of the method procedure.

---

### Official Review · Reviewer_2Y36 · 2024-11-04

**Soundness:** 3
**Presentation:** 3
**Contribution:** 1
**Rating:** 3
**Confidence:** 4

**Summary:**

The paper introduces Zoomer, a visual prompting mechanism designed to improve the performance of multimodal large language models (MLLMs) on vision-language tasks. MLLMs often struggle with visual data due to token limits, leading to omitted critical information and reduced accuracy. Zoomer addresses these challenges through three key innovations: a prompt-aware strategy that dynamically highlights relevant image areas, a spatial-preserving orchestration schema that maintains object integrity, and a budget-aware prompting method balancing global context with essential visual details. The mechanism's effectiveness is demonstrated through comprehensive evaluations across multiple datasets, showing that Zoomer consistently outperforms baseline methods with up to a 26.9% improvement in accuracy and significant reduction in token usage.

**Strengths:**

1. The idea sounds and easy to follow.
2. I appreciate the author draw Figure 3 for readers to better understand the framework.

**Weaknesses:**

> ### 1. Limited Novelty

- Hilight the important subregion to help the MLLM better answer the question is not a new idea. Simliar ideas can be found in [1] VStar [2] DualFocus.
- This paper doesn't propose any new innovatitive modules, but employing several existed techniques or modules like GroundingDINO and NMS.

> ### 2. Incomplete Benchmarks

- Only benchmark close-source models but without open-source MLLMs like Qwen-VL or LLaVA
- Not benchmark general MLLM benchmarks like SEED, MMBench

> ### 3. Cascade system, error propagation

- Zoomers relies on the GroundingDINO on the first stage, the error arised from the GroundingDION (wrongly localize objects) may affects the MLLM in the second stage.


[1] V*: Guided Visual Search as a Core Mechanism in Multimodal LLMs

[2] DualFocus: Integrating Macro and Micro Perspectives in Multi-modal Large Language Models

**Questions:**

Q1: Zoomer depends on the external grounding model GroundingDINO, which is good at grounding objects given explicit description or category name. However, for some implicit objects that needs reasoning, GroundingDINO may be hard to localize the specific area. For example, assume there is an image where there are three disks, two are on the desk and the other one is on the floor. But the question only asks how many disks are on the desk like, "How many disks are on the desk", Will GroundingDINO only ground two disks on the desk?

Q2: How about Zoomer on open-source MLLMs like Qwen-VL and LLaVA, and general MLLM bencharks like MMBench and SEED?

Q3: What if GroundingDINO ground wrong objects?

---

### Official Review · Reviewer_9W1y · 2024-11-04

**Soundness:** 3
**Presentation:** 2
**Contribution:** 3
**Rating:** 3
**Confidence:** 4

**Summary:**

The paper titled Zoomer: Enhancing MLLM Performance with Adaptive Image Focus Optimization addresses the limitations of current multimodal large language models (MLLMs) in visual processing, specifically in high-detail recognition and token-constrained environments. The authors propose Zoomer, a novel visual prompting mechanism aimed at preserving critical visual details within strict token limits. Zoomer introduces three main components: a prompt-aware strategy to dynamically highlight relevant image regions, a spatial-preserving orchestration to maintain the spatial integrity of objects, and a budget-aware prompting method that balances global context with essential visual details. Experimental results across multiple datasets show that Zoomer significantly improves accuracy—up to 26.9% on some datasets—while reducing token usage by as much as 67%, proving its efficiency and adaptability for black-box MLLMs.

**Strengths:**

1. The paper presents an innovative approach to improving visual processing in MLLMs without requiring architectural changes, which is particularly relevant given the constraints of black-box models.

2. The paper is mostly well-structured and provides sufficient background to understand the challenges it addresses.

3. The empirical evaluations covering a range of datasets and black-box models which have demonstrated Zoomer's efficacy in various visual and multimodal tasks.

**Weaknesses:**

1. Although the paper mentions related methods, it lacks in-depth comparative analysis with similar approaches in the literature. A stronger contextualization of Zoomer’s contribution relative to specific existing techniques, such as visual search [1], would provide readers with a clearer understanding of its potential advantages.

2. The ablation study could benefit from more comprehensive analysis. Specifically, breaking down the impact of individual modules within Zoomer on different datasets would clarify the relative importance of each component, such as the prompt-aware visual emphasizing versus the budget-aware strategy.

3. The paper would benefit from a deeper analysis of its computational costs and latency, in addition, the authors should provide more experiments about adopting object detection models, such as using different object detection models to determine its robustness or on datasets with varying levels of object detection difficulty.

[1] Penghao Wu, Saining Xie, "V*: Guided Visual Search as a Core Mechanism in Multimodal LLMs".

**Questions:**

1. Could the authors elaborate on how the spatial-preserving orchestration schema maintains the relative positions of image slices? Additional details about this aspect, potentially with illustrative examples, would clarify how Zoomer handles complex layouts in high-resolution images.

2. How does Zoomer compare with existing visual prompting mechanisms that employ patch-based approaches in black-box MLLMs? While some comparisons are made, a direct benchmark against well-known methods in similar contexts would strengthen the results.

---

### Official Review · Reviewer_7ZAN · 2024-11-06

**Soundness:** 2
**Presentation:** 3
**Contribution:** 2
**Rating:** 5
**Confidence:** 4

**Summary:**

This work proposes Zoomer, a visual prompting mechanism for multimodal large language models (MLLMs). Current MLLMs struggle with visual data processing due to unified strategies and token limits. To this end, Zoomer has three innovations: a prompt-aware strategy highlighting relevant image regions, a spatial-preserving schema maintaining object integrity, and a budget-aware prompting method. Evaluations on multiple datasets show Zoomer outperforming baseline methods, with up to 26.9% accuracy improvement and reduced token consumption, thus enhancing MLLM performance in vision-language tasks.

**Strengths:**

1. The paper presents an interesting solution to addressing the limitations of multimodal large language models (MLLMs) in processing visual data. The proposed Zoomer offers a perspective on how to enhance the performance of MLLMs in vision-language tasks, as well as considering the challenges posed by token limits and the need to preserve visual details.

2. The research methodology is reasonable. The authors conduct pilot experiments to identify the issues with current MLLM image processing strategies. They use a variety of datasets (such as Vstar, CVBench, and RealworldQA) and black-box MLLMs (like GPT - 4o, Gemini Pro, and Claude 3) for evaluation. The ablation study further helps in understanding the contributions of individual components within Zoomer.

3. The paper is well-structured and easy to follow. The introduction effectively outlines the problem of MLLMs in handling visual data and the limitations of existing approaches. The description of each component of Zoomer - the prompt-aware visual emphasizer, spatial-preserving orchestration schema, and budget-aware prompting strategy - is detailed. The experimental setup, including the datasets used, models evaluated, and metrics considered, is clearly explained.

4. The work has potential implications for the field of artificial intelligence, particularly in the area of multimodal learning. By improving the performance of MLLMs in vision-language tasks, it can have applications in various domains such as image captioning, object recognition, and interactive question-answering systems. The ability to preserve visual details within token limits is crucial for real-world applications where computational resources are limited.

**Weaknesses:**

1. The paper focuses on improving existing MLLMs through a visual prompting mechanism. However, it could have explored the potential of alternative model architectures that might better handle visual data.

2. Although the paper evaluates on multiple datasets, some of the more complex visual scenarios could have been explored in more detail. For example, in real-world applications, there are often occlusions, multiple objects of interest with complex relationships, etc. The performance of Zoomer in such highly complex and variable visual situations could have been analyzed more thoroughly to better understand its limitations and potential for improvement.

3. The idea in this paper is very similar to TextCoT [1]. The authors should cite TextCoT, discuss the difference from it, and justify the superiority over it with quantitative comparison.
[1] Bozhi Luan, Hao Feng, Hong Chen, Yonghui Wang, Wengang Zhou, Houqiang Li. TextCoT: Zoom In for Enhanced Multimodal Text-Rich Image Understanding. arXiv:2404.09797.

4. The authors are encouraged to show some failure cases qualitatively and discuss the potential assumption under which the proposed method works.

**Questions:**

Please refer to the comment above.

---

### Note · Authors · 2024-11-15

I have read and agree with the venue's withdrawal policy on behalf of myself and my co-authors.